# Assessment of Quality of Life after Endovascular and Open Abdominal Aortic Aneurysm Repair: A Retrospective Single-Center Study

**DOI:** 10.3390/jcm11113017

**Published:** 2022-05-27

**Authors:** Johanna Gruel, Eberhard Grambow, Malte Weinrich, Thomas Heller, Justus Groß, Matthias Leuchter, Mark Philipp

**Affiliations:** 1Department of Otorhinolaryngology, Head and Neck Surgery “Otto Körner”, Rostock University Medical Center, 18057 Rostock, Germany; 2Department of General, Visceral, Thoracic, Vascular and Transplantation Surgery, Rostock University Medical Center, 18057 Rostock, Germany; eberhard.grambow@med.uni-rostock.de (E.G.); justus.gross@med.uni-rostock.de (J.G.); matthias.leuchter@med.uni-rostock.de (M.L.); mark.philipp@med.uni-rostock.de (M.P.); 3Department for Vascular Medicine, DRK Kliniken Berlin Köpenick, 12559 Berlin, Germany; m.weinrich@drk-kliniken-berlin.de; 4Institute of Diagnostic and Interventional Radiology, Pediatric Radiology and Neuroradiology, Rostock University Medical Center, 18057 Rostock, Germany; thomas.heller@med.uni-rostock.de

**Keywords:** aortic aneurysm, vascular surgery, incisional hernia, EVAR, OR quality of life

## Abstract

Postoperative quality of life is an important outcome parameter after treatment of abdominal aortic aneurysms. The aim of this retrospective single-center study was to assess and compare the health-related quality of life (HRQoL) of patients after open repair (OR) or endovascular treatment (EVAR), and furthermore to investigate the effect of incisional hernia (IH) formation on HRQoL. Patients who underwent OR or EVAR for treatment of an abdominal aortic aneurysm between 2008 and 2016 at a University Medical Center were included. HRQoL was assessed using the SF-36 questionnaire. The incidence of IH was recorded from patient files and by telephone contact. SF-36 scores of 83 patients (OR: *n* = 36; EVAR: *n* = 47) were obtained. The mean follow-up period was 7.1 years. When comparing HRQoL between OR and EVAR, patients in both groups scored higher in one of the eight categories of the SF36 questionnaires. The incidence of IH after OR was 30.6%. In patients with postoperative IH, HRQoL was significantly reduced in the dimensions “physical functioning”, “role physical” and “role emotional” of the SF-36. Based on this data, it can be concluded that neither OR nor EVAR supply a significant advantage regarding HRQoL. In contrast, the occurrence of IH has a relevant impact on the HRQoL of patients after OR.

## 1. Introduction

Cardiovascular diseases are currently the leading cause of death worldwide [1]. Abdominal aortic aneurysms (AAA), along with vascular pathologies such as peripheral artery disease (PAD) and stenoses of the carotid artery, are among the five most frequent vascular diagnoses in German hospitals [2]. Treatment of AAA includes open repair (OR) or endovascular aortic repair (EVAR). Both procedures are considered equivalent with the respective benefits of each technique [3]. While endoleak formation is a specific complication after EVAR [4], one of the most important specific complications after OR is the development of an incisional hernia (IH). The incidence of IH in AAA patients varies from 10 to 38% [5,6,7], while studies with a longer follow-up period even showed incidences of up to 69% after OR [8,9]. Risk factors for IH include the AAA itself, obesity, malnutrition, corticosteroid medication, connective tissue disease, smoking, and pulmonary diseases such as chronic obstructive pulmonary disease (COPD), bronchial asthma, or chronic cough [10,11,12,13]. IH, in turn, often require secondary surgery. To prevent the development of IH after OR, the current European Society for Vascular Surgery (ESVS) guidelines on the management of abdominal aorto-iliac artery aneurysms as well as the German S3-guideline on screening, diagnosis, therapy, and follow-up of AAA, recommend considering prophylactic mesh reinforcement after median laparotomy for patients at high risk for IH [3,14]. In contrast, the Society for Vascular Surgery (SVS) practice guidelines on the care of patients with an abdominal aortic aneurysm and the British National Institute for Health and Care Excellence (NICE) guideline on ‘Abdominal aortic aneurysm: diagnosis and management’ do not state any recommendations regarding prophylactic mesh reinforcement [15,16].

To evaluate the benefit of aortic aneurysm repair, the health-related quality of life (HRQoL) became an important parameter in recent years [17,18,19]. The reason for this is that most procedures are performed electively on asymptomatic patients. Furthermore, the quality of life after therapy is as important as the technical success.

The aim of this study was to evaluate the HRQoL of patients after OR and EVAR. Furthermore, the effect of IH on HRQoL was investigated within the OR group.

## 2. Materials and Methods

This retrospective cohort study was approved by the local ethics committee (A2020-0168). All patients were included who underwent either OR or EVAR due to an AAA at the study site, a University Medical Center, between 2008 and 2016. The local database SAP (Walldorf, Germany) was screened for the OPS codes 5-38a, 5-384.5, 5-384.6, and 5-384.7 of the German Operationen- und Prozedurenschlüssel (version 2020). Additionally, digital and analog record search was conducted.

### 2.1. Assessed Data

Documented data included age, gender, body mass index (BMI), preexisting hernia, and concomitant diseases (coronary artery disease (CAD), PAD, COPD, smoking, chronic kidney disease, hyperlipidemia (HLP) and diabetes) at the time of OR or EVAR. In addition, data including the performed operation or intervention, whether it was emergency or elective treatment, and the incidence of postoperative complications, including IH and the respective management, were studied. All data were documented with Excel (Microsoft Excel 2019, Microsoft, Redmond, WA, USA).

### 2.2. Quality of Life Assessment

For the survey of patients’ HRQoL, the 36-item short-form health survey (SF-36) was used. This health questionnaire consists of 36 questions that reflect eight domains. These are physical functioning (PF), role physical (RP), bodily pain (BP), general health (GH), vitality (V), social functioning (SF), role emotional (RE), and mental health (MH) [20].

Due to contact restrictions at the time of data collection patients were contacted by phone only. After consenting to take part, patients received the SF-36 by post and were asked about their current health status and the occurrence of IH by phone. Subsequently, the questionnaire and a written consent form were completed and returned by post.

### 2.3. Aortic Access in OR and Routine Follow-Up

As intraoperative access route median laparotomy was performed in 72.2%, retroperitoneal access in 16.7%, and initial laparoscopy in 11.1%. In all cases, fascia closure was performed in standardized continuous fashion using two running CT-1 PDS (polydioxanone) loop sutures (Ethicon^TM^, Raritan, NJ, USA) in the small bites technique. No mesh was primarily applied for hernia prevention. For follow-up after surgery, patients were offered annual readmissions to our vascular outpatient clinic, which included an ultrasound examination of the aorta and femoral/popliteal arteries in addition to clinical examination.

### 2.4. Conduction of EVAR and Routine Follow-Up

Strictly following our internal institutional standard all EVAR-procedures have been performed either in the hybrid operation theater or in a dedicated cath lab under general anesthesia with open surgical access in both groins by an interdisciplinary team of anesthesiologists, interventional radiologists, and vascular surgeons. Depending on anatomical characteristics of the aneurysm and extension to the common iliac arteries the following endograft systems were used: Excluder^®^ (W. L. Gore and Associates, Flagstaff, AZ, USA) in 20/47 cases, Endurant II^TM^ (Medtronic, Dublin, Ireland) in 21/47 cases, Zenith Flex^®^ (Cook Medical Inc, Bloomington, IN, USA) in 5/47 cases and Ovation^TM^ (Endologix, Irvin, CA, USA) in 1/47 cases. After the intervention, all patients are transferred to the recovery room for monitoring. 

Routine follow-up included CTA before discharge as well as 6 resp. at 12 months after EVAR. This was followed by annual control by means of sonography or CTA depending on aneurysm size, kidney function, and sonographic assessability of the aorta.

### 2.5. Statistics

Program R was used for statistical analysis [21]. Both groups, OR and EVAR, were examined for differences within the above-mentioned parameters. When comparison of the groups involved whole numbers in terms of patient numbers (*n*), Fisher’s exact test was used to determine significance. Data of each subscale were compared between the groups using the Students’ t-test. Data from the SF-36 were compared using Mood’s median test. Binary logistic regression was performed to examine the effect of certain patient characteristics on the incidence of IH. Selected characteristics were sex, age, whether the surgery was elective or emergent, nicotine abuse, BMI, HLP, CAD, diabetes, and preoperatively existing hernia. *p*-values < 0.05 were considered significant.

## 3. Results

Out of the initially identified 258 patients treated for an AAA, 175 cases had to be excluded due to missing contact data, failed contact, death, inability, or denial to participate. In total, questionnaires of 83 patients were available, of whom 36 were treated by OR and 47 by EVAR (Figure 1).

### 3.1. Patient Characteristics

The mean age was 64.0 ± 8.8 years in the OR group at the time of operation and significantly lower compared to 70.2 ± 6.9 years in the EVAR group (*p* < 0.05). The groups also showed a significant difference in frequency of current smoking (OR: 27.8%, EVAR: 19.1%, *p* < 0.05) and prevalence of diabetes (OR: 30.5%, EVAR: 10.6%, *p* < 0.05). Arterial hypertension was the most common concomitant disease in both groups (OR: 91.7%, EVAR: 72.3%, *p* < 0.05). Other concomitant diseases, gender distribution, and mean follow-up did not differ significantly between the groups (Table 1).

For elective AAA repair (*n* = 76) EVAR (*n* = 46/47) was performed significantly more often compared to OR (*n* = 30/36, *p* < 0.05). For emergent interventions (*n* = 7) OR (*n* = 6/36) was performed significantly more frequent (EVAR: *n* = 1/47, *p* < 0.05).

### 3.2. Complications after EVAR and OR

After EVAR, the most frequent complication was the occurrence of endoleaks (21.3%) in ten patients, nine of whom underwent successfully revised endovascularly. Type I endoleaks were found in four patients (8.5%), and type II endoleaks in six patients (12.8%).

Other complications included one case of AAA rupture 8 years after intervention (2.1%), one thrombotic occlusion of a prosthetic leg 19 months after EVAR (2.1%), stenosis of a prosthetic leg in three cases (6.4%), and one inguinal wound infection (2.1%). Each of these complications required at least one reintervention. No complications occurred in 32 patients (68.1%).

The most common early postoperative complication after OR was postoperative bleeding. One patient (2.8%) suffered postoperative bleeding as well as abdominal compartment syndrome. Two patients (5.6%) had isolated postoperative bleeding. One patient developed colonic ischemia (2.8%). Relaparotomy was necessary in each of these four cases (11.2%). No complication occurred in 32 patients (88.9%).

Three patients (8.3%) in OR had a history of abdominal wall hernia prior to AAA detection (two umbilical, one inguinal hernia). Two of them developed an IH after OR. Overall, eleven patients (30.6%) in the OR group developed an IH. Three of them (27.3%) were treated by mesh-based hernioplasty in sublay technique. None of the characteristics gender (*p* = 0.19), age (*p* = 0.75), elective or emergency operation (*p* = 0.87), current smoking (*p* = 0.35), BMI </≥30 (*p* = 0.49), HLP (*p* = 0.63), CAD (*p* = 0.66), diabetes (*p* = 0.67), and preexisting hernia (*p* = 0.17) had a significant impact on IH development in the studied population.

### 3.3. HRQoL after EVAR and OR

The average follow-up time between the first intervention (OR or EVAR) and the survey of the SF-36 questionnaire was 7.1 years (OR: 7.8 years, EVAR: 6.6 years). The time ranged from a minimum of 3.6 years (OR and EVAR) to a maximum of 12.4 years (OR: 12.1 years, EVAR: 12.4 years).

Significantly higher values in the SF-36 were found in the OR group in the domain MH (OR: 62 (60–64) compared to EVAR: 60 (60–60), *p* < 0.05) (Figure 2). In the domain PF, there was no significant difference, but a trend with higher scores after EVAR (EVAR: 80 (55–90) compared to OR: 65 (43.75–85), *p* = 0.05). In the other six domains, no significant difference was detected.

### 3.4. Effects of IH on HRQoL after OR

A significant difference was found in three of eight domains of the questionnaire between HRQoL of patients with and without IH (Figure 3). Significantly higher values were obtained by patients without IH in the domains PF (IH: 65 (40–65) compared to w/o IH: 70 (45–90), *p* < 0.05), RP (IH: 25 (0–100) compared to w/o IH: 100 (25–100), *p* < 0.05) and RE (IH: 100 (100–100) compared to w/o IH: 100 (100–100), *p* < 0.05).

## 4. Discussion

This retrospective single-center analysis found that the choice of surgical procedure for AAA repair did not differ in terms of HRQoL in EVAR and OR. In contrast, in patients after OR developing an IH the HRQoL is affected significantly.

In sense of patient-centered medicine, recording the effect of a specific therapy on patients is crucial [22]. A suitable method is the analysis of quality of life as subjectively experienced health from the patient’s point of view [23]. With decreased morbidity and mortality in various treatment techniques, HRQoL became a more important parameter in clinical trials [18,24,25]. The SF-36 is currently the most widely used and fast performable questionnaire for surveying HRQoL [26]. Its reliability has been confirmed in several studies [27,28,29]. Answering the questionnaire takes an average of ten minutes. Due to its validity and reliability, the SF-36 has been recommended as the preferred questionnaire instrument for patients with vascular disease [29]. However, HRQoL is not an objectively measurable variable [18]. Still, in a long-term follow-up design of a retrospective study, it seems crucial to include HRQoL because reliable outcome parameters are scarce.

In the present study, a significant difference was only found in the domain MH of the SF-36 with lower values in patients after EVAR. In comparison, the EVAR group in the prospective DREAM study reached significantly higher HRQoL values than the OR group in the SF-36 domains PF, SF, and RP at three weeks postoperatively. In contrast, after twelve months, patients in the OR group revealed significantly higher scores in the domains PF, SF, RE, BP, and GH [30]. In the EVAR-1-trial, the early postoperative period of up to three months also showed slightly higher HRQoL in the EVAR group compared to the OR group. However, the results of the HRQoL survey after one to two years revealed no significant difference between the two groups [31]. The need for continuous follow-up due to the risk of endoleaks in EVAR patients, requiring additional interventions, was discussed as a factor affecting HRQoL in the DREAM study [18]. In contrast, no evidence was found in the EVAR-1-trial that post-interventional monitoring affected HRQoL after EVAR [31]. In the present study, all patients were recommended annual follow-up after surgery by physical examination and sonography. In comparison, endovascularly treated patients received a CTA within five days after EVAR. In the case of type II endoleak not requiring therapy immediately, CTA was repeated after six months. If no endoleak was detected, routine follow-up was performed by CTA 12 months after EVAR, followed by annual controls by sonography or CTA. Although the effects of follow-up assessments were not specifically investigated in this study, the more frequent CT examinations for endoleak detection after EVAR could have a long-term effect on quality of life, leading to the significantly lower scores in the domain MH in patients after EVAR. 

High rates of endoleaks requiring close follow-up assessments and additional re-interventions are significantly more frequent when endovascular therapy is not performed within the manufacturer’s instructions for use (IFU) of implanted stent-graft [32]. Thereby the most frequent reasons why patients are ineligible for EVAR are too short aortic neck length and too small diameters of the distal aorta and the iliac access arteries. In order to extend endovascular therapies for a broader range of aorto-iliac anatomies, new low-profile endografts have been developed [33]. Like the Ovation, these endostent grafts allow EVAR in patients whose vascular anatomy does not meet the IFU criteria of previous stent grafts. Therefore, it might be assumed that more frequent use of these low-profile devices, which are even eligible for aortic necks with a diameter of 0.7 mm, could lead to fewer endoleaks and in turn to fewer re-interventions. In this context, Zavatta et al. recently showed that the use of the Incraft (Cordis Corporation, Bridgewater, NJ) ultra-low-profile endograft revealed both sufficient technical success rates and freedom from re-interventions [34]. Similar positive results were recently reported in a retrospective analysis of the low profile endografts Ovation, Zenith LP, and Incraft [35]. This could also positively affect patients’ quality of life.

In addition to follow-ups and re-interventions, the vascular access performed for EVAR also affects the quality of life. Surgical access to the common femoral artery was performed for all EVARs in our study. Although percutaneous access was shown to be beneficial on complication rates, procedure time and hospital length of stay [36,37], its effect on patients’ quality of life has not been extensively studied yet. However, it might be assumed that although percutaneous access could improve quality of life, this effect might only be evident in the early post-interventional phase and not in a long-term follow-up.

There are only a few studies with a follow-up time of more than one year. In the DREAM study, the HRQoL of patients was collected over 5 years, in another study the median was 5.2 years [15,29]. In the present work, the median time between treatment and collection of the SF-36 was 7.8 years after OR and 6.5 years after EVAR. As a limitation, it should be noted that due to the study design, the time between treatment and SF-36 assessment varied among patients, limiting comparability with data from prospective studies. Thus, the minimum time span was 3.6 years for both OR and EVAR and the maximum was 12.1 and 12.4 years for OR and EVAR, respectively.

Another important aspect is that all the patients studied suffer from other diseases besides abdominal aortic aneurysm, which could additionally affect the HRQoL studied. However, the comparison performed on concomitant diseases between the two groups revealed no differences except for the diagnosis of diabetes, arterial hypertension, and smoking, which were significantly more frequent in the OR group. Since this comparison is based on preoperative findings, it is possible that patients might have developed other concomitant diseases postoperatively. Due to the retrospective design of our study, no SF-36 survey before treatment is available to show a baseline HRQoL.

In this study, the incidence of complications and respective surgical or endovascular revisions differed markedly between EVAR and OR. Thereby both complications and revisions were more frequent after EVAR. A significantly higher proportion of complications and reinterventions were also observed in the EVAR-1-trial after EVAR. Here, 41% of the patients after EVAR had at least one complication, compared to 9% after OR. At least one reintervention was performed in 20% after EVAR but only in 6% after OR [31]. The lower complication and reintervention rate after OR in our study reflects this benefit of OR. However, the assessed HRQoL does not reflect this advantage relevantly.

No statistically significant difference was found with respect to gender distribution between EVAR and OR. However, patients in the EVAR group had a significantly higher mean age. This is in line with the general recommendation of national and international guidelines to prefer OR in younger healthier patients [3,14].

The occurrence of IH is usually assessed and documented during clinical follow-up. In our department, all patients after EVAR and OR are offered annual follow-up. However, only 63.9% of patients after OR attended such appointments. Therefore, it is still unclear whether the incidence of IH recorded in this study corresponds to the actual incidence. Although all patients were also asked about the occurrence of IH during the phone contact, it remains unclear, whether the patient-side assessment was correct. It is known that more than 30% of patients with IH were not aware of their presence. This observation involved especially older patients whose IH was small [38].

In our study, 30.6% developed an IH after OR. Comparable results were found by Musella et al., who found an IH of 31.7% after OR was [6]. In contrast, van Ramshorst et al. described an incidence of IH of only 20%. However, the median follow-up period of 1.3 years was short in this study [39]. In contrast, markedly higher incidences after OR were observed in two other studies with 59.4% [9] and 69.1% [8].

While most IH occurs within the first two years after surgery [40], the majority of IH develops in the first three postoperative years [39,40]. Consequently, a corresponding long-term follow-up is necessary [10,39,41]. In our study, the median duration after OR was 7.8 years. Due to the retrospective study design, the shortest period was 3.6 years after OR in two patients. However, even this time should have been sufficient to detect an existing hernia [42].

Several risk factors are associated with IH. In our study, no significant influence on the appearance of IH was found for any of the studied diseases. Another retrospective study demonstrated that AAA patients with a BMI above 25 had a significantly higher risk for IH than patients with a BMI below 25. Both male sex and positive smoking history were not significantly associated with the occurrence of IH, as this was also found in the present study.

We found, a significant influence on HRQoL in patients with IH after OR in the three domains PF, RP, and RE of the SF-36. In the PRIMA trial, after a two-year follow-up period, HRQoL assessed by the SF-36 revealed no significant differences between patients with and without IH after median laparotomy in seven domains. Only in the domain PF, do patients without IH have significantly higher scores. In addition, patients with IH had significantly higher scores on the visual analog score for postoperative pain [43]. In another study, addressing the effect of IH on HRQoL and body image after open abdominal surgery by means of SF-36, after one year, patients with IH had significantly lower scores in the domains PF and RP than patients without IH [39]. Symptoms caused by IH range from aesthetic discomfort and skin problems to mild discomfort due to abdominal and lower back pain, constipation, pulmonary dysfunction, limitations in daily and work life, and serious complications such as obstruction, incarceration, and strangulation [39,44,45,46,47]. In contrast, half of the patients with IH are completely asymptomatic [38]. This circumstance might explain the unaffected domains of the SF-36 questionnaire in patients with IH in the present work.

Muysoms et al. found a highly significant reduction of IH after OR when a mesh was implanted compared to conventional fascia closure. After two years, the incidence of IH was 28% after conventional closure versus 0% after mesh insertion [48]. The AIDA trial also demonstrated a significant reduction in the incidence of IH with prophylactic mesh insertion. One year after OR, the incidence was 4.55% in the mesh group and 21.74% in the conventional closure group [49]. These results show that prophylactic use of mesh insertions after OR is an effective way to prevent IH.

Other studies reported mesh infections in almost 20% after hernioplasty [48,49]. However, data specific to the preventive use of mesh show that the rate of surgical wound infections does not differ significantly between patients with and without mesh insertion [50,51,52]. In the PRIMA study, the only disadvantage of mesh insertion was the incidence of intrabdominal abscesses, which occurred more frequently in the mesh group [53].

## 5. Conclusions

In summary, we found that occurrence of IH after OR revealed a significant impact on HRQoL. Further scientific evidence is needed to evaluate the impact of IH after OR on HRQoL more comprehensively. With a view on the ongoing development of low-profile and ultra-low-profile devices in recent years, studies on the HRQoL of these new, promising devices are needed to assess their impact on patients’ quality of life. For this purpose, prospective studies with larger patient cohorts are needed. In the context of the present work, the importance of a continuous follow-up regarding both vascular complications and the documentation of late complications, such as IH, must be emphasized.

## Figures and Tables

**Figure 1 jcm-11-03017-f001:**
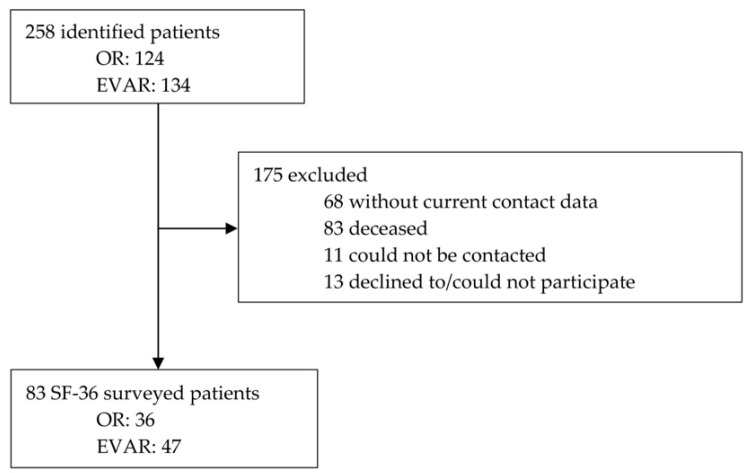
Trial profile. From 2008 to 2016, 258 patients were treated with open or endovascular repair at the University Medical Center. After exclusion of 175 cases, 83 patients were included in the study. Health-related quality of life was assessed by means of the SF-36 questionnaire. OR: open repair; EVAR: endovascular aortic repair.

**Figure 2 jcm-11-03017-f002:**
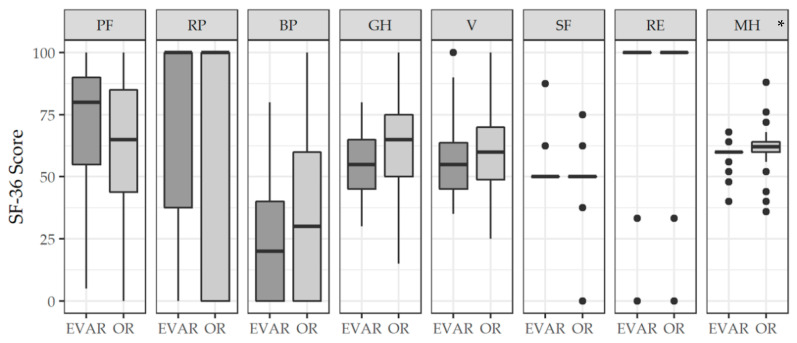
Health-related quality of life in patients after endovascular (EVAR) and open repair (OR) of an abdominal aortic aneurysm was assessed by means of the SF-36 questionnaire. Data are given as median and IQR (25% and 75% percentile). Dots represent suspected outliers (≥1.5 IQR). Mood’s median test. * *p* < 0.05 vs. EVAR. IQR: interquartile range; PF: physical functioning; RP: role physical; BP: bodily pain; GH: general health; V: vitality; SF: social functioning; RE: role emotional; MH: mental health.

**Figure 3 jcm-11-03017-f003:**
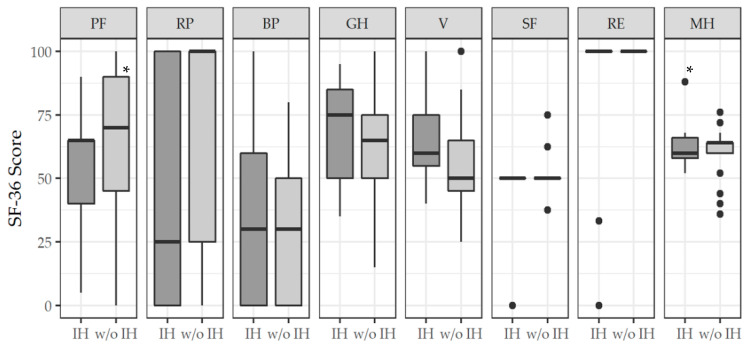
SF-36 score in patients after open repair of an abdominal aortic aneurysm. The health-related quality of life was assessed depending on the occurrence of incisional hernia (IH) compared to patients without incisional hernia (w/o IH), respectively. Data are given as median and IQR (25% and 75% percentile). Dots represent suspected outliers (≥1.5 IQR). Mood’s median test. * *p* < 0.05 vs. IH. IQR: interquartile range; PF: physical functioning; RP: role physical; BP: bodily pain; GH: general health; V: vitality; SF: social functioning; RE: role emotional; MH: mental health.

**Table 1 jcm-11-03017-t001:** Baseline characteristics at the time of data collection and follow-up time of patients after open repair or endovascular aortic repair of an abdominal aortic aneurysm. Data are given as *n* (%) or mean ± SD. Fisher’s exact test.

	OR *n* = 36		EVAR *n* = 47	
	*n*	%	*n*	%
Age	64 ± 8.8		70.2 ± 6.9 *	
Male	32	88.9	45	95.7
Female	4	11.1	2	4.3
CAD	14	38.9	19	40.4
Arterial hypertension	33	91.7	34	72.3 *
HLP	18	50.0	16	34.0
Smoking	10	27.8	9	19.1 *
Diabetes	11	30.5	5	10.6 *
COPD	6	16.7	6	12.8
BMI	28.01 ± 3.62		29.29 ± 3.05	
Elective intervention	30	83.3	46	97.9 *
Emergent intervention	6	16.6	1	2.1 *
Follow-up (years)	7.8 ± 2.7		6.6 ± 1.9	
Min. follow-up (years)	3.6		3.6	
Max. follow-up (years)	12.1		12.4	

* *p* < 0.05 vs. OR. SD: Standard deviation; OR: open repair; EVAR: endovascular aortic repair; CAD: coronary artery disease; HLP: hyperlipoproteinemia; COPD: chronic obstructive pulmonary disease; BMI: body mass index.

## Data Availability

The data presented in this study are available on request from the corresponding author. The data are not publicly available.

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
