# Peer review of "Assessment of Quality of Life after Endovascular and Open Abdominal Aortic Aneurysm Repair: A Retrospective Single-Center Study"

_jcm, 2022, doi:10.3390/jcm11113017_

Round 1

Reviewer 1 Report

A retrospective cohort study on the impacts of the endovascular and open AAA repair was carried out in this paper. This study's major result was that IH significantly impacts HRQoL among patients who underwent OR. The study was well designed and carefully performed.

Author Response

The authors thank the reviewer for the comments and the review of the manuscript.

Reviewer 2 Report

Topic of average interest, but my main concerns regard the few cases collected and analyzed in order to support the conclusions of the authors. The  sample of patients in the two cohorts of EVAR vs OS should be increased significatively.

Author Response

The authors thank the reviewers for the comments and the constructive criticism to improve the content of the paper. We appreciate this and have followed and clarified all comments in detail. Changes made in the manuscript were highlighted with underlining and red font.

  • Topic of average interest, but my main concerns regard the few cases collected and analyzed in order to support the conclusions of the authors. The sample of patients in the two cohorts of EVAR vs OS should be increased significatively.

The authors thank the reviewer for his comment. Indeed, a larger sample size would be desirable to strengthen the significance of our results and to obtain more reliable data. However, the number studied patients was limited by different aspects.

First, this is a single centre analysis. So, the number of treated patients in our department is the limits the case that could be studied.

Second, the retrospective study design. Although 258 patients were treated for abdominal aortic aneurysm between 2008 and 2016, only 83 patients could be included in the retrospective analysis because contact information were missing or incorrect, because patients were unable or unwilling to participate in the study, or because they had already died.

Third, the minimal time between the operation and assessment of health-related quality of life was determined at 3 years because we wanted to study the potential effect of incisional hernia on quality of life. Since most incisional hernia develop within 3 years after laparotomy, it is actually not possible to include patients who were treated after 2019. However, it would be possible to contact patients who were treated between 2017 and 2019, which would increase the number in each group by approximately 10-20 cases per group. However, this prolonged analysis would take more than the given 10 days.

An assessment of the quality of life of patients treated before 2008 does not appear to be meaningful, since it can be assumed that numerous patients have already died in the course of time or cannot be reached due to incorrect or missing contact data. Thus, this approach does not seem to us to be effective in increasing the number of cases in a sufficient way. 

As proposed in the conclusion (please see page 9, line 338-356), a prospective multi-centre study is needed to address the issue of health-related quality of life after abdominal aortic aneurysm repair in a sufficiently with a meaningful number of studied cases.

However, if the reviewer is convinced that the additional assessment of patients treated between 2017 and 2019 is crucial, the authors would request 4 weeks more to contact the respective patients by telephone and by mail.

Reviewer 3 Report

The aim of the paper was to retrospectively assess the quality of life after endovascular and open abdominal aortic aneurysm.  The topic of the paper is interesting but there are some elements that have to be clarified.

One of the most confounding points is that the Authors reserved a lot of attention to the influence of Incisional hernia on quality of life after AAA repair. From the title, I wouldn't expect such interest in this specific topic. In this light, I would suggest better organising the discussion section. 

I would suggest also implementing the EVAR-dedicated parts because I found that EVAR is less discussed with respect to OR.

Some crucial limitations are already addressed by the Authors, the retrospective nature of the study and the lack of a pre-operative HRQoL.

In the Materials and Methods section, I would suggest the Authors add a paragraph dedicated to EVAR procedures. There is no data regarding the type of procedure performed (anaesthesia, surgical or percutaneous access, standard or low-profile endograft...). This is quite important for the readers because endovascular treatment changed a lot during the past decade. Thanks to technological development, indications for EVAR have become wider and wider addressing a lot of initial limitations. In this light, I would suggest adding some words regarding EVAR and how has changed in order also to reduce the procedural stress (percutaneous access, locoregional anaesthesia, low-profile endograft....) allowing to treat also elderly patients. 

Please add the typical follow-up scheme that has been used. This could be of help, especially regarding the possible impact that follow-up could have on the quality of life. 

Results are clear, but again, the space dedicated to IH is overexpressed in relation to what the title offer. 

Some references that could help:

doi: 10.1016/j.jvs.2021.09.036. Epub 2021 Oct 9. PMID: 34634415.

doi: 10.1080/17434440.2017.1281738

doi: 10.1016/j.jvs.2020.09.044.

Author Response

The authors thank the reviewers for the comments and the constructive criticism to improve the content of the paper. We appreciate this and have followed and clarified all comments in detail. Changes made in the manuscript were highlighted with underlining and red font.

  • One of the most confounding points is that the Authors reserved a lot of attention to the influence of Incisional hernia on quality of life after AAA repair. From the title, I wouldn't expect such interest in this specific topic. In this light, I would suggest better organising the discussion section.

The authors agree with the reviewer. The discussion was critically revised and EVAR is now discussed in more detail according to the reviewers’ suggestions and the given literature. In line with this, some parts on IH were removed (please see the revised discussion).

  • I would suggest also implementing the EVAR-dedicated parts because I found that EVAR is less discussed with respect to OR.

The authors agree with the reviewer. EVAR is now discussed in more detail, especially the potential benefits of low profile endografts. Due to the limited word count, the part of OR discussion was shortened (please see the revised discussion).

  • In the Materials and Methods section, I would suggest the Authors add a paragraph dedicated to EVAR procedures. There is no data regarding the type of procedure performed (anaesthesia, surgical or percutaneous access, standard or low-profile endograft...). This is quite important for the readers because endovascular treatment changed a lot during the past decade. Thanks to technological development, indications for EVAR have become wider and wider addressing a lot of initial limitations. In this light, I would suggest adding some words regarding EVAR and how has changed in order also to reduce the procedural stress (percutaneous access, locoregional anaesthesia, low-profile endograft....) allowing to treat also elderly patients.

The authors completely agree with the reviewer. In order to follow the reviewers advice, we now mention the used endografts as well as the performed anesthesia and vascular access in methods and discuss these aspects as well (please see page 3, line 94-107 and page 6-7, line 225-256).

  • Please add the typical follow-up scheme that has been used. This could be of help, especially regarding the possible impact that follow-up could have on the quality of life.

The authors thank the reviewer for this comment. The typical follow-up schemes in our department after open and endovascular aortic aneurysm repair are now included in the methods section (please see page 2 and 3, line 90-93 and 105-107). For a better interpretation of the impact of follow-up on the studied quality of life, the follow-up schemes were also included in the discussion (please see page 6, line 225-227).

  • Results are clear, but again, the space dedicated to IH is overexpressed in relation to what the title offer.

In order to follow the reviewer’s comment, the discussion was revised. EVAR is now more pronounced than incisional hernia to the reduce the overexpression of IH in the discussion. However, the initial discussion of incisional hernia, including frequency, risk factors, effects on quality of life and the possible prevention by mesh implantation was so intense because our data suggest that this has an important impact on the quality of life, which was, regarding this retrospective single-centre analysis, more pronounced than the treatment by means of OR or EVAR.

The authors kindly thank the reviewer for the provided literature on low-profile endovascular stent grafts which we now included and discussed in the manuscript (please see revised discussion).

Round 2

Reviewer 2 Report

I still maintain my perplexities on the size of the population examined in this study.
This is a work well designed and well set up, but it need a much wider sample of patient that could be reached only with a Multicentric study.

Reviewer 3 Report

The paper has been considerably improved after the first round of revision. Nevertheless, some elements remained unclear, especially regarding the overall study design.